# The Cause and Effect Relationship of Diabetes after Acute Pancreatitis

**DOI:** 10.3390/biomedicines11030667

**Published:** 2023-02-22

**Authors:** Mariola Śliwińska-Mossoń, Iwona Bil-Lula, Grzegorz Marek

**Affiliations:** 1Department of Medical Laboratory Diagnostics, Division of Clinical Chemistry and Laboratory Haematology, Wroclaw Medical University, Borowska 211A, 50-556 Wroclaw, Poland; 2Grzegorz Marek (GM) Second Department and Clinic of General and Oncological Surgery, Wrocław Medical University, Borowska 213, 50-556 Wrocław, Poland

**Keywords:** acute pancreatitis, post-pancreatitis diabetes mellitus, diabetes type 3c, pancreatogenic diabetes, pancreatic exocrine insufficiency

## Abstract

Acute pancreatitis (AP) is an acute inflammation of the pancreas associated with high morbidity and mortality. Endocrine pancreatic insufficiency secondary to AP has drawn increasing attention in recent years. The aim of this paper is to analyze the available clinical and experimental literature to determine the cause and effect relationship of diabetes type 3c (T3cDM; pancreatogenic diabetes) after acute pancreatitis. The clinico-pathological features and management challenges of pancreatogenic diabetes overlap with other secondary causes of diabetes. A complex pathogenesis involving pancreatic exocrine insufficiency, dysfunction of insulin secretion, and insulin resistance is likely the cause of T3cDM after AP. To obtain an improved understanding of the pathophysiology of diabetes after AP, more research is now needed to understand the risk of complications related to the pancreas and diabetes in these patients.

## 1. Introduction

Acute pancreatitis (AP) causes organ damage associated with a risk of death [1,2,3]. AP has always been considered a self-limiting disease, in many cases reversible, but with the current state of knowledge about AP, it is now also known to be the cause of glucose metabolism disorders [4]. The results of an extensive meta-analysis of recent literature have shown that prediabetes and/or diabetes (DM) affect almost 40% of patients after AP [5]. In addition, it was found that patients with diabetes after an attack of acute pancreatitis have a higher risk of death and readmission to hospital than patients with type 2 diabetes [6].

According to the latest guidelines [7], the development of diabetes after acute pancreatitis is classified as exocrine pancreatic diabetes. These results have changed the generally stated opinion that morphological changes in the pancreas are reversible and that the organ’s endocrine function fully recovers after treatment of acute pancreatitis, regardless of severity [8,9,10]. DM can develop after an AP attack, but there is disagreement about the factors and extent of the risk of endocrine pancreatic insufficiency secondary to acute pancreatitis [5,11,12,13]. The underlying mechanisms by which DM develops after AP have not been unreservedly elucidated. Due to the increased risk of morbidity and mortality resulting from the coexistence of AP and DM, it is important to clarify the role of AP in the mechanism of DM onset and/or progression. Clarification of the role of AP in the development of DM may provide information for the development of preventive strategies in this group of patients. The aim of this paper is to analyze the clinical and experimental literature to investigate the cause and effect relationship of diabetes after acute pancreatitis.

## 2. Acute Pancreatitis

Acute pancreatitis is one of the most common acute conditions in gastroenterology. In 15–20% of patients, AP presents as severe inflammation, and complications can be life-threatening and are associated with 5% mortality [3]. The frequency of acute pancreatitis is rising throughout the world. The rise is partly due to increases in metabolic syndrome and hypertriglyceridemia. The peak incidence of acute pancreatitis is in the fifth and sixth decades of life; however, there is an increase in mortality with age. In the United States, population incidence has been cited as 600 to 700 per 100,000 people, with 200,000 to 250,000 discharges occurring yearly for acute pancreatitis [14].

Localized destruction in the pancreas and a systemic inflammatory response characterizes the pathophysiology of pancreatitis. The major inciting event is the premature activation of the enzyme trypsinogen to trypsin within the acinar cell instead of the duct lumen. This premature activation of these zymogens causes extensive tissue damage and recruiting neutrophils, which initiates the inflammatory cascade. This inflammatory cascade then leads to the systemic manifestations of acute pancreatitis [3,14].

According to the revised Atlanta classification, acute pancreatitis can be diagnosed using the criteria in Table 1.

Since in about a quarter of patients the first two criteria alone may not indicate acute pancreatitis, caution is required. Additionally, it is possible that one in ten patients with acute pancreatitis will be misdiagnosed [16]. It is known that the activity of pancreatic enzymes (lipase or amylase) on admission to the hospital does not correlate with the severity of the disease [17].

In practice, imaging studies are rarely conducted to establish a diagnosis of AP; instead, a diagnosis is most often made on the basis of clinical symptoms and biochemical tests. The onset of acute pancreatitis is considered to be the time when pain occurs, not the time of admission to the hospital [3,15].

Alcohol and gallstones account for a total of 80% of AP, and idiopathic AP accounts for 10%, though this may mostly be the result of undiagnosed gall microlithiasis. There have been relatively few studies of drug-associated acute pancreatitis. Trivedi and Pitchumoni as well as Badalov et al. have described four categories of drugs that may induce acute pancreatitis or pancreatic injury [18,19]. Forty-three drugs met the criteria to be included in class I, including codeine, metronidazole, tetracycline, valproic acid, and hydrocortisone. The following medications have met the criteria to be included in class II: acetaminophen, α-methyldopa, mesalamine, all-trans-retinoic acid, and valproate. The data implicating the drugs in class III are weaker than those for the previous two classes. These drugs include alendronate, captopril, carbamazepine, metformin, and naproxen. Many single-case reports exist in the literatures that describe the association of various medications in class IV with the development of acute pancreatitis. Drugs included in this class include diclofenac, ketoprofen, nitrofurantoin, and ritonovir [19]. In addition, it should be remembered that although there are effective strategies to reduce the risk of developing acute pancreatitis after endoscopic retrograde cholangiopancreatography (ERCP), ERCP is a recognized factor in AP development [20,21]. The causes of acute pancreatitis are presented in Table 2.

There are also many rarer risk factors for AP that clinicians must consider to address potential complications and/or prevent recurrence, such as: (a) trauma; (b) hypercalcemia; (c) viral infections; (d) cancers; (e) anatomical variants; (f) heart bypass surgery; and (g) organophosphate poisoning [23,24,25].

Acute pancreatitis can be subdivided into two types: interstitial oedematous pancreatitis and necrotizing pancreatitis [15] (Figure 1). Interstitial oedematous pancreatitis affects the majority of patients and is characterized by a generalized form or local enlargement of the pancreas due to inflammatory oedema and at most minor inflammatory changes in the tissues surrounding the pancreas. It is mild and usually resolves within a week. Necrotizing pancreatitis, found in 5–10% of patients, is diagnosed on the basis of a lack of tissue enhancement after administration of a contrast agent in computed tomography. Necrosis develops within a few days, so in order to avoid underestimating its extent, computed tomography should be performed optimally after about 5–7 days from the onset of the disease. Necrosis usually affects both the pancreas and the peripancreatic tissues, but sometimes it affects only the pancreas or only its surroundings. In the further course of the disease, the necrosis may remain solid or fluid, or may become resorbed or infected [15,26].

The clinical severity of acute pancreatitis is stratified into three categories: mild, moderately severe, and severe [3] (Figure 2).

The underlying mechanisms of inflammation and necrosis are not fully understood, but it has generally been recognized that activated leukocytes play a significant role in the pathogenesis of inflammation. Serum concentrations of pro-inflammatory cytokines, such as tumor necrosis factor α, and interleukins, such as interleukin-1 beta (IL-1), interleukin-6 (IL-6), and interleukin-8 (IL-8), were significantly higher in severe pancreatitis compared with mild pancreatitis [27,28].

During recovery from an attack of acute pancreatitis, both exocrine and endocrine transient organ failure may occur [29,30]. Due to the possibility of pancreatic insufficiency, pancreatic function should be monitored for up to three months after an attack of acute pancreatitis. Endocrine pancreatic function should be monitored after about three months (by fasting and postprandial blood sugar concentrations, and possibly by HbA1C measurements) [3,31]. The above studies are necessary as acute pancreatitis has been found to be a factor in accelerating diabetes of the exocrine pancreas, and especially as there is a higher background prevalence of AP than any other pancreatic disease. A large epidemiological study showed that AP was the most common disease of the pancreas preceding diabetes [32]. Another study of the New Zealand population found a high percentage of AP (61%) causing T3cDM, and the results of this study also showed a high overall prevalence of 1.13 per 1000 population [33]. Additionally, a large systematic review and meta-analysis revealed a high frequency (15%) of T3cDM in the first year after AP onset, with a 23% increase over the next 5 years, and then an increase to 40% in the subgroup with a follow-up of more than 5 years. There was a statistically significant 2.7-fold increased risk of newly diagnosed DM at 5 years or more in comparison with that at 12 months and below [5].

## 3. The Type 3c Diabetes Mellitus

Diabetes is a group of metabolic diseases characterized by chronic hyperglycemia resulting from impaired insulin secretion and/or action. Insufficient insulin secretion and/or reduced tissue response to insulin impairs the complex action of insulin in target tissues, resulting in impaired carbohydrate, lipid, and protein metabolism. Both insulin secretion and function may be impaired. Diabetes can also develop in the course of other diseases, including diseases of the exocrine part of the pancreas. The current classification of diabetes mellitus comprises diabetes mellitus types 1–4 [7]. Whereas nearly every physician is aware of the existence of type 1 and type 2 DM, diabetes mellitus secondary to pancreatic diseases (type 3c, pancreatogenic diabetes) is a condition seldom thought of in everyday clinical practice. In recent years, there has been evidence to suggest that type 3c diabetes may be more common than previously thought. Studies have shown that this type of diabetes can be consistently underdiagnosed and misdiagnosed [34].

Based on the above data, the prevalence of diabetes in the general population is estimated to be approximately 6.6% worldwide, and according to the latest estimates, it will continue to increase [35]. Whereas the prevalence of type 1 and type 2 diabetes is well characterized, data on type 3c diabetes mellitus are not adequate.

Type 3c is often called brittle diabetes because glucose control is especially challenging absent an appropriate beta cell (insulin) or alpha cell (glucagon) response. Patients with type 3c diabetes are more likely to experience complications and death related to hypoglycemic events [36].

A recent study has compared pancreatogenic diabetes with type 2 diabetes. Pancreatogenic diabetes was characterized by a lower body mass index (BMI), required more insulin, had more episodes of hypoglycemia, but had lower rates of dyslipidemia and hypertension. It was found that T3cDM is characterized by a low BMI and the absence of elements of the metabolic syndrome [37].

Pancreatogenic diabetes as a group is associated with an impaired incretin effect. It has been shown that in fibrocalculous pancreatic diabetes, basal C-peptide and glucagon are low, and glucagon concentration is not increased after glucose load. Glucagon-like peptide 1 (but not gastric inhibitory peptide) increased 1.5 to 2 times in fibrocalculous pancreatic diabetes as compared with T2DM and controls (fasting and post glucose), without differences in dipeptidyl peptidase IV [38].

T3cDM is associated with an incidence ranging from 25% to 80% with chronic pancreatitis [35,36,37,38,39,40,41,42,43]. The risk of diabetes in patients with chronic pancreatitis increases with the duration of the disease and with the severity of damage to the pancreas, especially in the early form of pancreatic calcifications. The second most common cause of T3cDM (8% of all T3cDM) seems to be pancreatic cancer [35,43]. It also rises with a prior distal pancreatectomy [31].

Studies have determined that during fasting, pancreatic polypeptide concentrations and responses to meal stimulation are reduced in pancreatogenic diabetes associated with pancreatic ductal adenocarcinoma or chronic pancreatitis compared with T2DM [43].

Recent studies have shown that 60.2% of adult patients developed DM after AP episodes, and 9% of children after acute recurrent pancreatitis or with chronic pancreatitis, and this risk likely increases as they age into adulthood [5,11,44]. Several studies have also been published on the prevention and treatment of newly diagnosed diabetes after AP, but risk factors remain controversial [3,9,45]. Therefore, a mechanism of T3cDM has been proposed, including inflammation, fibrosis, and hardening of the endocrine tissue of the pancreas, which leads to a gradual reduction and loss of β-cell function, leading to impaired or non-insulin secretion [35,36,37,38,39,40,41,42] (Figure 3).

There are also disturbances in the remaining pancreatic islet cells and a change in the secretion of glucagon, somatostatin, and pancreatic polypeptide [46], which have consequences for the clinical condition of the patient. In summary, T3cDM affects all Langerhans islet cells and therefore has features of both insulin resistance and insulin deficiency. Additionally, it was found that the patients are at risk of a hyperglycemia and a hypoglycemia event with an increased insulin requirement in the early stage of the disease [47].

A diagnosis of pancreatogenic diabetes was and still is difficult to make. The basic diagnostic criteria include laboratory tests such as those performed when type 2 diabetes is suspected, including clinical symptoms of hyperglycaemia and glucose ≥ 200 mg/dL (11.1 mmol/L), or fasting plasma glucose (fasting is defined as no caloric intake for at least 8 h) with at least two biochemical abnormalities, e.g., ≥126 mg/dL (7.0 mmol/L), or 2-h glucose (PG) ≥ 200 mg/dL (11.1 mmol/L) after a 75 g oral glucose tolerance test, or hemoglobin A1c (HbA1c) ≥ 6.5% (48 mmol/mol). Generally, fasting plasma glucose, 2-h glucose after a 75 g oral glucose tolerance test, and HbA1c are equally appropriate for diagnostic screening, and one of the above criteria is sufficient to diagnose diabetes [7].

The diverse etiology of T3cDM associated with acute or chronic pancreatic inflammation, but also with pancreatic cancer, cystic fibrosis, pancreatic surgical resections, and autoimmune pancreatitis, causes the disease development to be variable and heterogeneous. Therefore, a determination of the role of pancreatic dysfunction in the diagnosis of T3cDM remains open and dependent on the underlying disease of this organ [48,49]. In addition, the pathophysiology of pancreatogenic diabetes is multifactorial, with different potential contributions in a given patient resulting from (a) loss of mass of pancreatic islet cells, especially beta cells; (b) autoimmunity; (c) local and systemic inflammatory response; (d) mutations of the CFTR (CF transmembrane conductance regulator) proteins; (e) deficiency of fat-soluble vitamins A, D, E, and K; and (f) disruption of the insulin-incretin axis (nesidioblastosis) [47,48,49,50,51] (Figure 3 and Figure 4).

The question remains whether all diabetes associated with pancreatic disease should be considered as T3cDM. More stringent diagnostic standards were proposed by Ewald and Bretzel [26], but these were criticized because they are difficult to implement clinically [49,50,51,52]. Plain abdominal ultrasound can play an important role in diagnosing nonalcoholic pancreatogenic diabetes [53]. The proposed criteria of the researchers are presented in Table 3 [34,54,55].

The diagnosis of T3cDM is definitely complicated, and the diagnosis is sometimes ambiguous, as long-term patients with type 1 and type 2 diabetes are associated with exocrine pancreatic insufficiency [50] and patients with diabetes are more likely to develop acute and/or chronic pancreatitis [35,40]. Based on the results of four large retrospective studies, it was found that type 2 diabetes increases the risk of acute pancreatitis by 1.86–2.89 times [56,57,58,59]. Additionally, it has been established that patients who previously experienced attacks of acute pancreatitis may develop type 1 or type 2 diabetes, regardless of exocrine pancreatic dysfunction. Moreover, adipose tissue, the gastro-intestinal tract, and kidney function may also be involved in disease development [60]. It has been observed that exocrine hypofunction of the pancreas develops in parallel with the dysfunction of the endocrine function of the organ [61].

Most recently, a trend of positive correlations between high-density lipoprotein cholesterol and amylase in T2DM patients has been discovered [62]. Increased hyperlipidemia and oxidative stress are some of the major triggers for conditions such as atherosclerosis, diabetic angiopathy, thrombotic events, and cardiovascular problems, all of which are complications of diabetes [63].

A recent study on gut microbial dysbiosis in CP observed a unique bacterial signature in the gut of patients with type 3c diabetes that differs from that observed in type 1 and type 2 diabetes [64]. It is known that the pathogeneses of the three types of diabetes differ significantly. Type 1 and type 2 diabetes are associated with insulin deficiency resulting from autoimmunity and insulin resistance, respectively, while type 3c diabetes is associated with a combination of insulin deficiency and hepatic insulin resistance [31,39,65]. Additionally, Sasikala et al. found that type 3c diabetes was associated with islet inflammation and Th17 cell infiltration, resulting in IFN-γ secretion, which causes a functional defect in beta cells [66]. It has also been suggested that isletitis may be caused by endotoxemia resulting from intestinal microbial dysbiosis and altered permeability of the intestinal mucosa in these patients [64]. Another cross-sectional study reported significant differences in the gut microbiome in patients with type 3c diabetes compared with those with type 1 and type 2 diabetes, confirming the involvement of the microbiota in the pathogenesis of pancreatogenic diabetes [67].

In general, there is no evidence-based treatment for type 3c diabetes, which is uniquely different from type 1 and type 2 diabetes, and treatment should be based on cause and pathophysiology [36]. Treatment of type 3c diabetes mellitus should include treatment of endocrine and exocrine pancreatic insufficiency. Insulin is the most commonly prescribed therapy for patients with pancreatogenic diabetes associated with advanced chronic pancreatitis. However, there is no specific insulin regimen [68]. Caution is required in dosing and monitoring patients with type 3c diabetes on insulin therapy because of the risk of hypoglycaemia due to impaired glucagon function. There is some evidence to support the use of metformin as a first-line therapy in patients with T3c diabetes. This drug has several important effects: metformin may reduce the amount of insulin needed daily to lower blood glucose levels; it can protect against the development of pancreatic ductal adenocarcinoma [39,69]; it may cause an increase in GLP-1 levels and additionally increased expression of pancreatic GLP-1 and GIP receptor genes (studies conducted on mice) [70]; and it may prolong the survival of diabetic patients without pancreatic cancer metastases (32% lower risk of death) [71]. In addition to the drugs mentioned above, the following drugs have been used in the treatment of endocrine dysfunction: suphonylureas, thiazolidinediones, DPP-4 inhibitors, GLP-1 analogues, and α-glucosidase inhibitors [55]. The management of exocrine pancreatic insufficiency in type 3c diabetes mellitus patients should include the administration of pancreatic enzymes to improve the absorption of fats and fat-soluble vitamins (particularly vitamin D supplementation). Pancreatic insufficiency causes the malabsorption of nutrients, and replacing pancreatic enzymes can improve nutrient absorption and incretin secretion, which ultimately improves glycemic control [36,72].

Long-term (up to 18 years of follow-up) studies to examine the risk of death and hospitalization in people with diabetes after pancreatitis (PPDM) compared with those with T2DM found that people with PPDM had a significantly higher mortality rate of 80.5 per 1000 person-years (95% CI, 70.3–90.6) compared with T2DM patients, who had a mortality rate of 1.13 per 1000 person-years (95% CI, 1.00–1.29). Additionally, patients with PPDM had a significantly lower mean age at death than subjects with T2DM (67.8 vs. 70.0 years, *p* < 0.001) [6]. In terms of cause, cardiovascular disorders were the most common cause of death in PPDM (mortality: 25.2 per 1000 person-years) [73].

In summary, individuals with PPDM have a higher risk of mortality (from cancer, infectious disease, and gastrointestinal disease) and hospitalization (for chronic pulmonary disease, moderate to severe renal disease, and infectious diseases) compared with individuals with T2DM [6]. The obtained results dictate the need to develop guidelines for the management of PPDM in order to prevent the excessive number of deaths and hospitalizations of diabetic patients after pancreatitis.

## 4. Clinical Severity of Acute Pancreatitis and the Development of T3cDM

In recent years, several studies on the relationship between the clinical severity of acute pancreatitis and the development of pancreatogenic diabetes have been published [9,45]. The severity of the AP disease depends mainly on the occurrence and extent of pancreatic necrosis, which reflects local pancreatic damage and the dysfunction of further organs, which in turn translates into systemic disorders [3].

In a study of Tu at al. to determine the effect of disease severity on the risk of developing new-onset diabetes after acute pancreatitis, the percentage of patients with pancreatic necrosis in the diabetic group was higher than that in the non-diabetic group, indicating a significant role of pancreatic necrosis in newly developed diabetes after AP [74]. Connor et al. [75] found that 33% of patients without prior diabetes who underwent a pancreatic necrosectomy developed endocrine disorders. Subsequently, Umapathy et al. [76] reported that symptoms of diabetes mellitus were observed in 45% of patients after acute necrotizing pancreatitis. This observation was confirmed by Bavare et al. [77], who stated that, following necrotic pancreatitis, pancreatic exocrine or endocrine disorders were observed in more than half of patients who had undergone a pancreatic necrosectomy. Tsiotos et al. [78] and Gasparoto et al. [79] also found that necrotizing pancreatitis had a significant effect on the disturbance of exocrine and endocrine pancreatic function in half of the patients in their studies. In these cases, pancreatic morphological changes were common (62.5%), and more so in patients with extensive necrosis. In another study, it was observed that pancreatic exocrine insufficiency strongly correlated with the extent of organ necrosis and pancreatic endocrine insufficiency [80]. Chandrasekaran et al. [81] showed a higher frequency of endocrine dysfunction in patients undergoing a necrosectomy (61.9% of patients compared with 28.5% of patients not undergoing surgery (*p* = 0.05)). Based on the above results, it can be concluded that necrotizing pancreatitis plays an important role in the development of diabetes. Extensive organ necrosis leads to atrophy or absence of pancreatic tissue, which results in a reduced number of normal β-cells of Langerhans islets and thus a reduced amount of secreted insulin [82]. It is a very similar mechanism to the pathogenesis of pancreatic diabetes after pancreatectomy [83,84,85,86]. Hamad et al. found that among 4255 patients who had undergone a pancreatoduodenectomy or distal pancreatectomy, with a median follow-up of 10.8 months after surgery, the incidence of pancreatogenic diabetes was 20.3%. In patients with at least 3 years of follow-up, diabetes developed in 32.2% [86].

In line with the necrosis–fibrosis hypothesis, using computed tomography, Avanesov et al. [87] found that the total pancreatic volume was significantly reduced in patients with recurrent AP compared with patients with a single episode of AP. In addition, significant endocrine and exocrine insufficiency was observed in patients with recurrent AP. The above data, mainly from severe cases of AP, suggest that the theory of increased islet cell loss due to pancreatic necrosis leads to a greater risk of developing T3cDM and impaired glucose metabolism. Additionally, a meta-analysis comparing the results of studies with severe and mild AP showed a higher incidence of diabetes after severe AP (39%) compared with mild AP (14%) [88].

AP-induced β-cell injury and the limited regenerative capacity of beta cells might account for pancreatic endocrine insufficiency. Based on studies performed in the AP mouse model as well as in pancreatic tissues from patients with acute necrotizing pancreatitis (ANP), a loss of β-cells was found in both ANP patients and AP mice. Additionally, activated cytokeratin 5 (Krt5+) positive pancreatic cells were observed in ANP pancreases along with persistently elevated Notch activity, resulting in massive duct-like structures. Notch signaling determines the differentiation of pancreatic progenitor cells in pancreatic development. AP mice that received the Notch inhibitor showed that impaired glucose tolerance was reversed 7 days and 15 days after AP, and the number of newborn small islets increased due to some extent to the increased differentiation of Krt5+ cells into beta cells. The results of the study confirm that a loss of β-cells contributes to the endocrine failure of the pancreas after AP, while inhibition of Notch activity promotes the differentiation of pancreatic Krt5 + cells into β-cells and improves glucose homeostasis [89].

However, there are also several studies in which the emergence of new diabetes has been associated with alcohol consumption, gender, and age, but has not been associated with the severity of acute pancreatitis [90,91]. In a meta-analysis involving a total of 24 prospective clinical trials of 1102 patients with their first episode of AP, Das et al. found that the severity of AP, its etiology, and the age and gender of the patients had a minimal impact on the development of prediabetes and diabetes [5].

If pancreatic β-cell loss is an obvious cause of DM after severe necrotic AP, then the pathogenesis of DM in mild AP is more complex and islet cell loss is not the only risk factor or mechanism. One theory is that AP triggers an immune response in patients genetically susceptible to the development of DM [5]. Antibodies against glutamine acid decarboxylase and islet cell antigen (IA2) have been detected in patients with latent autoimmune DM [92] and type 1 diabetes [93]. Another theory is that insulin resistance may play the dominant role in the development of T3cDM after AP [74]. In addition, the findings of the DORADO project have provided a wealth of information on the signaling molecules that influence and do not affect glucose metabolism in people with mild acute pancreatitis [94]. First, it was found that the early changes following mild acute pancreatitis include functional β-cell adaptation, which is best assessed by the disposition index. Second, the DORADO project established IL-6 as the primary mediator of cytokines in post-pancreatitis diabetes, which exerts its glucose-regulating effects through insulin resistance (an increase in IL-6 by 1 ng/mL was associated with a 0.7% increase in insulin resistance (*p* = 0.038)) [94,95]. It has been found that IL-6 causes impaired phosphorylation of the insulin receptor and insulin receptor substrate-1, leading to insulin resistance. A significant increase in IL-6 levels has been demonstrated in the course of acute pancreatitis, and it is possible that an early increase in IL-6 levels leads to chronic hyperglycemia after resolution of the acute disease [96,97,98]. In contrast, chronic hyperglycemia is a source of reactive oxygen species causing increased lipid peroxidation, which further increases the risk of DM (Figure 2).

Endothelin (ET) is a peptide secreted by the endothelial cells, vascular smooth muscle cells, macrophages, and the renal medulla [99,100]. ET-1 leads to structural changes in the pancreas, resulting in both exocrine and endocrine dysfunction, which cause pancreatic vasoconstriction, disrupting blood flow. ET-1 is considered a risk factor for pancreatic diseases, especially acute ischemia and pancreatitis, as it induces the secretion of pro-inflammatory cytokines (IL-1, IL-6), which exacerbate existing pancreatitis and lead to disease progression [101].

It has recently been suggested that metabolic factors such as obesity and hypertriglyceridemia put individuals at higher risk of developing AP, termed “metabolic AP” [102]. It has also been found that these factors may increase the risk of developing DM after AP [103]. It is generally known that obesity itself is a significant risk factor for hyperglycaemia and insulin resistance [104]. In a study by Mentula et al., it was noted that obesity may contribute to early hyperglycemia in AP patients. Additionally, a multivariate analysis showed that obesity is not an independent risk factor for organ failure. It has been shown that the occurrence of obesity in patients correlates with an early increase in blood glucose levels, which may predispose them to systemic disorders in the course of AP [105].

Other researchers have suggested that one of the possible mechanisms for the emergence of new diabetes after mild AP is associated with a deficiency of fat-soluble vitamins in patients with exocrine pancreatic dysfunction [106,107]. Vitamin D deficiency and lower serum levels of 25-hydroxyvitamin D occur in up to 94% of patients with exocrine pancreatic dysfunction [108,109], which is associated with a higher risk of DM [107,110].

The management of pancreatic diabetes after AP is difficult. In addition to routine diabetes care, including monitoring for nephropathy, retinopathy, and neuropathy, physicians must address complications related to pancreatitis, including post-pancreatitis pain, poor oral intake, psychological issues, exocrine pancreatic insufficiency, and vitamin and mineral deficiencies [111]. The management of pancreatic diabetes after AP includes behavioral/style interventions (e.g., abstinence from alcohol/substances, nutrition, pain control, exercise) and pharmacological therapies (e.g., pancreatic enzyme supplementation, insulin and oral hypoglycaemic therapies, and treatment of osteoporosis) [111].

## 5. The Impaired Glucose Homeostasis after AP

Data on the development of a new prediabetes or diabetes after an AP attack are inconclusive. Some reports suggest that glucose homeostasis returns completely to normal [8], while others indicate that its disturbance persists in a significant proportion of patients [112]. Moreover, the timing of the development of prediabetes and DM after an episode of AP remains unclear, as does the influence of the etiology and severity of AP. In a study written by Das et al. specifying the incidence of prediabetes and diabetes after the first attack of AP, it was found that prediabetes and/or DM were present in 37% of patients after AP. Newly diagnosed diabetes developed in 15% of patients within 12 months of their first AP attack, and the risk of diabetes was significantly increased after 5 years (relative risk 2.7 (95% CI 1.9 to 3.8)). It was found that 70% of patients required constant insulin therapy [5]. Glycemic control in patients with diabetes mellitus resulting from pancreatitis is more difficult than control in patients with T2DM. In a study conducted by Woodmansey et al. on a very large population, it was proved that the mean HbA1c levels were higher in people with diabetes following pancreatic disease than in people with type 2 diabetes. Post-pancreatic diabetes was associated with poor glycemic control (adjusted odds ratio 1.7 (1.3–2.2); *p* < 0.001) compared with type 2 diabetes, and insulin use over 5 years was 20.9% (14.6–28.9) with diabetes after acute pancreatitis. After analyzing the data, researchers concluded that exocrine pancreatic diabetes is often referred to as T2DM, but is characterized by poorer glycemic control and significantly greater insulin requirements [34].

Post-AP hyperglycemia is generally considered a transient event which almost completely resolves itself in almost all patients. It is now believed that fasting glucose should not be used in the diagnosis of diabetes during the course of pancreatitis (and arbitrarily within 90 days after hospitalization) because elevated glucose levels may reflect an acute stress response and may also be a consequence of pancreatitis treatment (e.g., parenteral nutrition, intravenous dextrose infusion) [113].

Petrov et al. proposed the use of the oral glucose tolerance test for the early diagnosis of diabetes because the test is not hampered by an acute stress response (as in the case of fasting glucose) and does not require a 90-day delay period (as in the case of glycated hemoglobin). The authors believe that the diagnosis of transient stress-induced hyperglycemia in the course of pancreatitis is important. However, stress hyperglycemia occurs in up to seven out of ten patients with pancreatitis, and indiscriminate blood glucose monitoring in most patients with pancreatitis during (or shortly after) hospitalization would not be helpful in diagnosing diabetes mellitus [97,113,114].

Insulin resistance may be a key mechanism in the development of diabetes after AP. In the conducted study, it was found that the HOMA-IR value during the observation in the DM group was higher than in the group without DM [76]. Balzano et al. [115] suggests that T3cDM is associated with the classic risk factors for type 2 diabetes, e.g., age, gender, family history of diabetes, and BMI, as well as β-cell dysfunction and insulin resistance, which emerged as important determinants. Pancreatogenic diabetes is a heterogeneous disease entity that largely overlaps with type 2 diabetes. However, the HOMA-IR value in the study was obtained during the follow-up phase, not during hospitalization. Recognition of T3cDM is important for the identification of underlying pancreatic disease that may require specific intervention. Insulin therapy is ultimately needed for most patients, and metformin should be considered when concomitant insulin resistance is present [54]. Hence, research is needed to confirm whether insulin resistance after acute pancreatitis is caused by AP.

## 6. Exocrine Pancreatic Dysfunction and the Development of T3cDM

Despite convincing evidence from in vivo and in vitro studies on the strict morphological and functional dependence between pancreatic acinar cells and Langerhans islet cells (the “insulin–acinar axis”) [116], the relationship between exocrine disorders and pancreatic endocrine disorders remains poorly characterized. Typically, clinical trials investigating the temporal relationship between pancreatic exocrine and endocrine dysfunction have been disproportionately focused on studying diabetes as a risk factor for exocrine pancreatic dysfunction (EPD). However, evidence for the inverse relationship, e.g., exocrine pancreatic dysfunction as a risk factor for the appearance of DM, was only circumstantial. The reason for the lack of evidence for the above thesis is the difficulty of determining the optimal study population due to the existence of multiple causes of exocrine pancreatic dysfunction, most of which do not subsequently result in a high incidence of new cases of DM. The only well-known example was cystic fibrosis, in which mutations in the CTFR gene occurred, causing exocrine pancreatic dysfunction and leading to cystic fibrosis-related diabetes. Studies from several recent systematic reviews of pancreatitis have shown that the incidence of EPD does not typically increase over time after acute pancreatitis, while the incidence of new onset diabetes is increasing [117,118,119]. In a cohort study, a higher incidence of diabetes was observed in patients with EPD (17.0%) than in patients without EPD (5.2%, (*p* < 0.001)). Additionally, the mean duration between EPD and the onset of DM was found to be 1.8 years [106].

In other studies, up to 40% of cases involve the occurrence of both endocrine and exocrine disorders after AP [45,91,108], but a small (3%) overlap of pancreatic dysfunction has also been shown [67]. The exocrine function measurements differed from one study to another. In a study by Tu et al. [120], which aimed to assess the incidence of endocrine and exocrine pancreatic insufficiency after AP, it was found that pancreatic exocrine insufficiency is more difficult to diagnose than endocrine insufficiency. Usually, symptoms such as abdominal pain, abdominal extension, and fatty diarrhea in combination with an X-ray and stool examination are used for an accurate diagnosis. In this study, 5.3% of patients suffered from abdominal pain, 10.6% of patients suffered from abdominal distension, and 15.04% of patients suffered from diarrhea after discharge from the hospital. Only 4.4% of the patients had a BMI less than 18 (kg/m^2^) [120]. The symptoms of exocrine pancreatic insufficiency are neither unusual nor specific, and are therefore of little diagnostic value. The FE-1 test was used to indirectly assess the exocrine function of the pancreas. It was found that 6.2% of patients could be diagnosed with severe exocrine pancreatic insufficiency (<100 μg/g) and that 29.2% of patients had only mild to moderate exocrine pancreatic insufficiency (100–200 μg/g) [120]. FE1 has been criticized as a test for exocrine function because some researchers doubt its specificity and sensitivity. Leeds et al. concluded that FE-1 < 100 µg/g is highly specific for pancreatic exocrine insufficiency, but that 100–200 µg/g may only offer limited specificity and sensitivity [121]. Additionally, when the pre-test probability of exocrine pancreatic insufficiency is low, it often leads to multiple false positives [122]. In a systematic review and meta-analysis by the COSMOS group, low serum amylase and lipase levels were significantly associated with type 2 diabetes, type 1 diabetes, excessive obesity, and metabolic syndrome [123]. However, the role of digestive enzymes in the pathogenesis of metabolic disorders requires further research. According to the definition of Ewald, T3cDM takes into account the need to prove exocrine dysfunction [34]. Therefore, the relationship between endocrine and exocrine disorders in AP patients should be determined, and exocrine pancreatic insufficiency should be assessed using FE-1 and strengthened with other diagnostic tools, such as magnetic resonance imaging of the pancreatic duct.

In addition, a population-based study has shown that pancreatic exocrine function (as measured by FE-1) is independently related to the composition and diversity of the gut microflora [124], confirming another possible mechanism underlying the association between EPD and new-onset diabetes [125]. Moreover, it is believed that altered intestinal microflora is a factor predisposing patients to insulin resistance and T2DM by intensifying inflammation [126,127]. Several studies have found an altered microflora of the small intestine in patients with exocrine pancreatic dysfunction [128,129]. The likely altered gut microflora in EPD patients may induce insulin resistance by promoting low-grade inflammation in the patient, leading to a new onset of diabetes mellitus. These data are in line with the results of the DORADO project, which found that low levels of inflammation were associated with levels of the pro-inflammatory cytokine IL-6 [94,95].

## 7. Conclusions

Acute pancreatitis carries a risk of developing diabetes after pancreatitis. Compared with T2DM, post-pancreatitis diabetes is associated with poor glycemic control, a much higher insulin requirement, a higher risk of developing cancer, death at an earlier age, and a significantly higher risk of death [111].

One of the possible causes of the increase in the number of complications and mortality associated with diabetes after AP is misdiagnosis in the early stages, leading to delayed diagnosis with undesirable complications. The current classification approach involves broadly categorizing patients as having AP-related DM based on the chronologic onset of DM in relation to the AP episode. The clinico-pathological features and management challenges of pancreatogenic diabetes overlap with other secondary causes of diabetes. A complex pathogenesis involving pancreatic exocrine insufficiency, dysfunction of insulin secretion, and insulin resistance is likely the cause of T3cDM after AP.

The consequences of the prompt and correct diagnosis of T3cDM after AP are far-reaching and include advising patients on diet and lifestyle, improving glycemic control, reducing short- and long-term complications, and increasing patient and physician satisfaction. There is hope that oxyntomodulin—an intestinal peptide involved in the regulation of pancreatic exocrine function—may function as a potential biomarker to differentiate DM after AP from T2DM [94].

Due to an improved understanding of the pathophysiology of DM after AP, more research is now needed on the effects of drug therapy on DM after AP. Further prospective studies will help us to understand the risks of complications related to the pancreas and diabetes in these patients.

## Figures and Tables

**Figure 1 biomedicines-11-00667-f001:**
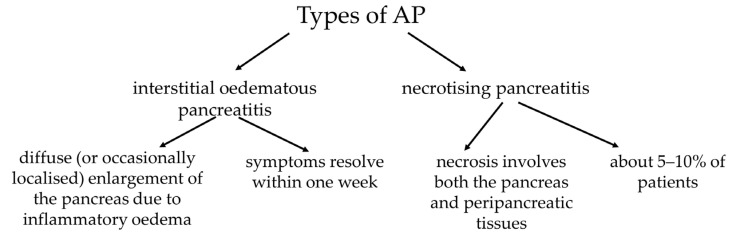
Types of acute pancreatitis [15,26].

**Figure 2 biomedicines-11-00667-f002:**
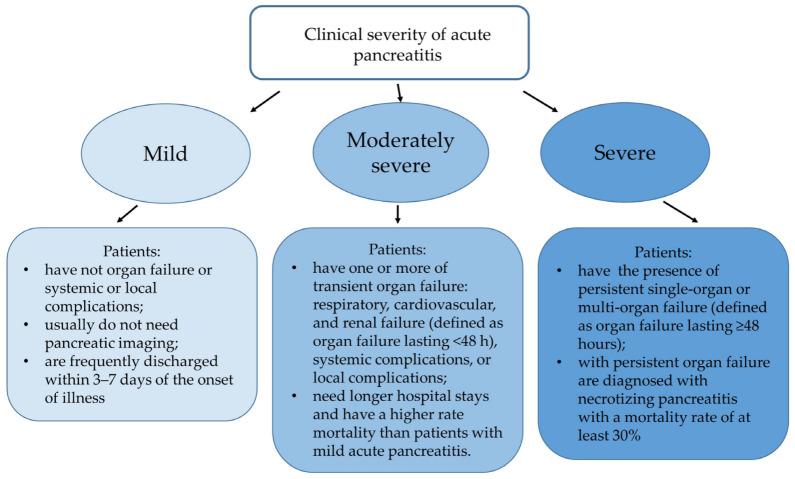
Clinical severity of acute pancreatitis [3].

**Figure 3 biomedicines-11-00667-f003:**
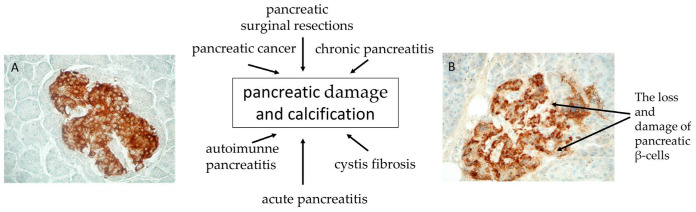
Immunohistochemical localization of insulin (**A**) in a normal pancreas and (**B**) in patients with diabetes type 3c. (**A**): (210×) Strong and very strong reaction of insulin in islet β-cells; (**B**): (265×) diffuse strong and moderate reaction of insulin in islet β-cells.

**Figure 4 biomedicines-11-00667-f004:**
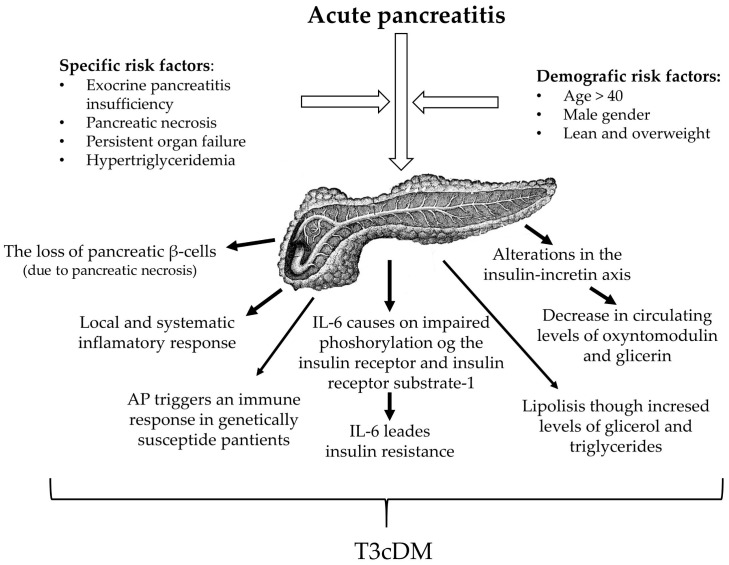
Proposed factors contributing to the pathophysiology of diabetes following acute pancreatitis, including potential mechanisms of dysglycemia.

**Table 1 biomedicines-11-00667-t001:** AP diagnosis criteria according to the Atlanta classification [15].

At Least Two of the Following Criteria Are Fulfilled	
1	Abdominal pain
2	An increase in serum lipase (or amylase) activity at least three times the upper limit of normal
3	Characteristic findings of acute pancreatitis in contrast-enhanced CT or, less often, MRI or transabdominal ultrasonography

**Table 2 biomedicines-11-00667-t002:** The causes of acute pancreatitis [22].

Frequent	Rare	Very Rare (<1%)
Alcohol Gallstones Idiopathic	Iatrogenic (after ERCP) Abdominal trauma Medications (e.g., azathioprine) Smoking	Hereditary pancreatitis Infection (viral, bacterial, parasitic) Hypercalcemia (hyperparathyroidism, excess vitamin D) Hypertriglyceridemia Birth defects (e.g., bifid pancreas) Sphincter of Oddi dysfunction Pancreatic tumors Poisons (e.g., scorpion venom) Vasculitis Autoimmune pancreatitis

**Table 3 biomedicines-11-00667-t003:** Criteria for diagnosing T3cDM proposed by Ewald and Bretzel [32,54,56].

Major Criteria (All Must Be Fulfilled):	
1	Evidence of exocrine pancreatic insufficiency (faeces elastase 1 (FE1) < 200 µg/g or incorrect direct function testing)
2	Pathological pancreatic imaging (endoscopic ultrasound, magnetic resonance imaging, and computed tomography)
3	Absence of type 1 diabetes mellitus-associated autoimmune markers.
**Minor Criteria:**	
1	Impaired beta cell function (e.g., HOMA-B, C-peptide/glucose-ratio)
2	No excessive insulin resistance (e.g., HOMA-IR)
3	Impaired incretin secretion (e.g., GLP-1, pancreatic polypeptide)
4	Low serum levels of lipid soluble vitamins (A, D, E, and K)

## Data Availability

The data presented in this study are available upon request from the corresponding author.

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
