# Peer review of "The Cause and Effect Relationship of Diabetes after Acute Pancreatitis"

_biomedicines, 2023, doi:10.3390/biomedicines11030667_

Round 1

Reviewer 1 Report

In this review, the authors summarized the epidemiology, diagnosis, pathological changes of islets,treatment about 3c diabetes after AP based on previous clinical literatures. The cause and effect relationship of diabetes after acute pancreatitis was emphasized. This topic is interesting, but several issues reduced the impact that would be expected of papers published in biomedicines. These concerns are listed below.

1) The structure of the manuscript is not clear, and some contents are lack of logic. Some chapters are divided into too many paragraphs and it is hard to understand the main idea. Such as the section “2. Acute pancreatitis”. The section“2. Acute pancreatitis“ is common sense contents that should be reduced to avoid lengthy.

2) The content of background is exact the same in abstract and introduction, which is inappropriate, please revise one of them.

3) Some important animal experimental data are not included in the manuscript, such as the pathological changes and electron microscope changes of islets after acute pancreatitis, please add these contents. (PMID: 34556631)

4) In line 71, page 2, what drugs can induce AP? I would suggest deleting the classification in line 80 and Figure 1.

5) The authors concluded to monitor pancreatic function for three month after pancreatitis, however the risk of diabetes after AP increases by 2 times after 5 years compared to 12 months. (PMID:23929695).

6) The figure 3A is from normal pancreatic specimen, the arrows may cause some confusion. Is figure 3b from type 2 diabetes or type 3c diabetes specimens?

7) Please use the three-line table to represent the tables.

8) Insulin resistance does exist in the early stages of acute pancreatitis. Conversely, peripheral insulin sensitivity can increase in type 3c diabetes (PMID:28507210).

9) The sentence “It is known that the prevalence of type 1 and type 2 diabetes is well characterized; type 3c are almost nonexistent”is confused.

10)  There are some minor grammatical errors, and inaccurate or incomplete citations. The authors need to reorganize the structure of the manuscript and carefully check the manuscript and cited references. 

Reviewer 2 Report

In this article, ÅšliwiÅ„ska-MossoÅ„ et al. review the relationship between diabetes (DM) and acute pancreatitis (AP). They discuss how acute pancreatitis, its severity, and subsequent exocrine dysfunction are often associated with the development of diabetes (referred to as T3cDM). Although there are a few excellent review articles published in recent years regarding the same topic, a few topics are uncommon in this article, resolving the novelty issue. Overall, the article is interesting as it covers an often-neglected pancreatic disorder and a “rare” type of DM. However, there are a few caveats that need to be addressed for the acceptance of the article.

  1. The article talks about acute pancreatitis; however, the pathophysiology, epidemiology, and sufficient background have not been provided. The authors can do that in section 2. A descriptive figure would be recommended as well.
  2. The article covers many topics; however, there are many instances where the point is incomplete. I am listing a few such examples, but the authors need to address this issue thoroughly throughout the article:
    1. Introduction section: What is the frequency of DM after AP and vice versa?
    2. Introduction or conclusion: What is the pathophysiological reason for increased morbidity and mortality with the coexistence of AP and DM?
    3. Section 2: The authors start the section by stating, “Acute pancreatitis is one of the most frequent acute conditions in gastroenterology.” How frequent is it?
    4. Section 2, Line 71: “Badalov et al. described four categories of drugs that may induce acute pancreatitis or pancreatic injury” What four categories of drugs? What is the significance? What is the mechanism of action of those drugs?
    5. Section 2, Line 76: “There are also many rarer risk factors for AP that clinicians must consider to address potential complications and/or prevent a recurrence, such as: (a) trauma, (b) hypercalcemia, (c) viral infections, (d) cancers, (e) anatomical variants, (f) heart bypass surgery, and (g) organophosphate poisoning [21-23].” How do these factors physiologically cause AP?
  3. Figure 1: A graphical figure would be better than a flow chart. Also, in the text description, the individual types of AP must be explained well.
  4. Figure 2: The color shading makes the reading difficult. A better combination of colors or shades must be picked up for easier reading.
  5. Figure 3: The labeling in this figure is confusing and needs to be clarified. While Figure 3A shows a normal pancreas, the labels showing different pancreatic exocrine diseases suggest that this is how the pancreas should look in these diseases.
  6. In lines 230 and 248, the authors have written “B” cell instead of pancreatic “β” cell. It needs to be corrected.
  7. The authors need to add a section on treatment strategies, current caveats (for diagnosis and treatment), and how those can be potentially addressed for AP and T3DM.
  8. The abstract and introduction start with the same four lines, and that too with glaring grammatical errors. These errors reduce the quality of the manuscript.
  9. The authors need professional proofreading to address many types of grammatical errors, including spelling, punctuation, sentence structures, and tenses.

Reviewer 3 Report

The paper entitled The Cause and Effect Relationship of Diabetes after Acute Pancreatitis” includes potentially relevant data for the implementation of individualized diagnostic and medical care in the fields of endocrinology, diabetology as well as for gastroenterology. Authors of this publication – among others – tend to suggest that “…Prompt and correct diagnosis of pancreatogenic disease after Acute Pancreatitis has far-reaching consequences…”.

However, I have a few remarks:

1. The text of the work lacks a commonly used synonym for Type 3c diabetes mellitus - pancreatogenic diabetes.

2. The article omits key publications from the last two years dealing with pancreatoginc disease (type 3c diabetes):

1. Hamad A, Hyer JM, Thayaparan V, Salahuddin A, Cloyd JM, Pawlik TM, Ejaz A. Pancreatogenic Diabetes after Partial Pancreatectomy: A Common and Understudied Cause of Morbidity. J Am Coll Surg. 2022 Dec 1;235(6):838-845. doi: 10.1097/XCS.0000000000000360.

2. Hart PA, Kudva YC, Yadav D, Andersen DK, Li Y, Toledo FGS, Wang F, Bellin MD, Bradley D, Brand RE, Cusi K, Fisher W, Mather K, Park WG, Saeed Z, Considine RV, Graham SC, Rinaudo JA, Serrano J, Goodarzi MO. A reduced pancreatic polypeptide response is associated with new onset pancreatogenic diabetes versus type 2 diabetes. J Clin Endocrinol Metab. 2022 Nov 21:dgac670. doi: 10.1210/clinem/dgac670.

3. Vonderau JS, Desai CS. Type 3c: Understanding pancreatogenic diabetes. JAAPA. 2022 Nov 1;35(11):20-24. doi: 10.1097/01.JAA.0000885140.47709.6f.

4. Amuedo S, Bellido V, Mangas Cruz MÁ, Gros Herguido N, López Gallardo G, Pérez Morales A, Soto Moreno A. Successful Use of an Advanced Hybrid Closed-loop System in a Patient With Type 3c Pancreatogenic Diabetes Secondary to Nesidioblastosis. Can J Diabetes. 2022 Sep 17:S1499-2671(22)00365-3. doi: 10.1016/j.jcjd.2022.09.117

5. Bellin MD. Pancreatogenic Diabetes in Children With Recurrent Acute and Chronic Pancreatitis: Risks, Screening, and Treatment (Mini-Review). Front Pediatr. 2022 Apr 26;10:884668. doi: 10.3389/fped.2022.884668. eCollection 2022.

6.  Valdez-Hernández P, Pérez-Díaz I, Soriano-Rios A, Gómez-Islas V, García-Fong K, Hernández-Calleros J, Uscanga-Dominguez L, Pelaez-Luna M. Pancreas. Pancreatogenic Diabetes, 2 Onset Forms and Lack of Metabolic Syndrome Components Differentiate It From Type 2 Diabetes. 2021 Nov-Dec 01;50(10):1376-1381. doi: 10.1097/MPA.0000000000001930.

7. Chakravarthy MD, Thangaraj P, Saraswathi S. Missed Case of Pancreatogenic Diabetes Diagnosed Using Ultrasound. J Med Ultrasound. 2021 Jan 9;29(3):218-220. doi: 10.4103/JMU.JMU_138_20. eCollection 2021 Jul-Sep.

8. Ghosh I, Mukhopadhyay P, Das K, Anne M B, Ali Mondal S, Basu M, Nargis T, Pandit K, Chakrabarti P, Ghosh S. Incretins in fibrocalculous pancreatic diabetes: A unique subtype of pancreatogenic diabetes. J Diabetes. 2021 Jun;13(6):506-511. doi: 10.1111/1753-0407.13139.

3. The Authors of the manuscript do not clearly address the issue of significant differences characterizing the gut microbiome of patients with pancreatogenic diabetes from gut microbiome of individuals suffering from Type 1 and Type 2 diabetes mellitus. The mentioned issue at least shortly needs to be described.

4. The Authors discussing the diagnostic criteria for type 3c diabetes refer to the article "Ewald N, Bretzel RG. Diabetes mellitus secondary to pancreatic diseases (Type 3c)--are we neglecting an important disease? Eur J Intern Med. 2013; 24:203-6. doi:10.1016/j.ejim.2012.12.017.".
However, other slightly "newer" publications, i.e.
"Andersen DK, Korc M, Petersen GM, Eibl G, Li D, Rickels MR, Chari ST, Abbruzzese JL. Diabetes, Pancreatogenic Diabetes, and Pancreatic Cancer. Diabetes. 2017 May;66(5):1103-1110. doi: 10.2337/db16-1477.;
Bhattamisra SK, Siang TC, Rong CY, Annan NC, Sean EHY, Xi LW, Lyn OS, Shan LH, Choudhury H, Pandey M, Gorain B. Type-3c Diabetes Mellitus, Diabetes of Exocrine Pancreas - An Update. Curr Diabetes Rev. 2019;15(5):382-394. doi: 10.2174/1573399815666190115145702." concerning the diagnostic criteria for type 3c diabetes (pancreatogenic diabetes) can be found.

5. Figures no. 1-4 are prepared carelessly, i.e. The font used in the figures is different than the one used in the text. All the Figures need to be prepared using the same font as the one used in the text. The resolution of the figures (no.3, no.4) is definitely not sufficient – The resolution of the figures (no.3, no.4) absolutely needs to be increased.

Round 2

Reviewer 2 Report

The authors have addressed the majority of my comments satisfactorily. The only minor concern that remains is the image quality. The authors need to make sure that they upload high-quality images.

Reviewer 3 Report

The authors made appropriate corrections. The article is ready for publication without any modifications.

Author Response

Dear Reviewer,

thank you very much for your time and review.

Best regards

Mariola Śliwińska-Mosson